# Assessment of In-Hospital Mortality and Its Risk Factors in Patients with Myocardial Infarction Considering the Logistical Aspects of the Treatment Process—A Single-Center, Retrospective, Observational Study

**DOI:** 10.3390/ijerph20043603

**Published:** 2023-02-17

**Authors:** Lukasz Gawinski, Monika Burzynska, Michal Marczak, Remigiusz Kozlowski

**Affiliations:** 1Department of Management and Logistics in Health Care, Medical University of Lodz, 90-237 Lodz, Poland; 2Department of Epidemiology and Biostatistics, Medical University of Lodz, 90-237 Lodz, Poland; 3Collegium of Management, WSB University in Warsaw, 03-204 Warsaw, Poland; 4Center of Security Technologies in Logistics, Faculty of Management, University of Lodz, 90-237 Lodz, Poland

**Keywords:** myocardial infarction, in-hospital death, logistical aspects of the treatment process

## Abstract

Technological progress, such as the launching of a new generation of drug-coated stents as well as new antiplatelet drugs, has resulted in the treatment of myocardial infarction (MI) becoming much more effective. The aim of this study was to assess in-hospital mortality and to conduct an assessment of risk factors relevant to the in-hospital death of patients with MI. This study was based on an observational hospital registry of patients with MI (ACS GRU registry). For the purpose of the statistical analysis of the risk factors of death, a univariate logistic regression model was applied. In-hospital general mortality amounted to 7.27%. A higher death risk was confirmed in the following cases: (1) serious adverse events (SAEs) that occurred during the procedure; (2) patients transferred from another department of a hospital (OR = 2.647, *p* = 0.0056); (3) primary percutaneous coronary angioplasty performed on weekdays between 10 p.m. and 8 a.m. (OR = 2.540, *p* = 0.0146). The influence of workload and operator experience on the risk of death in a patient with MI has not been confirmed. The results of this study indicate the increasing importance of new risk factors for in-hospital death in patients with MI, such as selected logistical aspects of the MI treatment process and individual SAEs.

## 1. Introduction

The dynamic development of medicine observed in the 21st century has resulted in a significant improvement in treatment results for the majority of medical problems and a decrease in mortality rates. Over recent years, cardiology has become a leader of this development. One of the best instances of the development of contemporary cardiology is the results from the treatment of coronary heart disease and, in particular, its final manifestation, namely, myocardial infarction (MI). In recent years, the values of both long- and short-term rates of mortality caused by MI have significantly decreased [1,2], which stem from several aspects. One of them is the enormous technological progress within the scope of invasive treatment. Modern materials, miniaturization, the development of invasive treatment methods, and the launching of a new generation of drug-eluting stents (DESs) and their widespread use have resulted in the interventional treatment of MI becoming very effective in recent years [3]. Another aspect is the development of contemporary pharmacotherapy. Launching new antiplatelet drugs to be applied in treatment has significantly improved the results of interventional treatment [4,5]. Furthermore, in terms of successful attempts to improve the relevant values connected with the treatment of MI, the operation of local health care systems has been perfected, a network of catheterization laboratories (CLs) has been launched, and health care workers have been working round the clock, backed up by telematic systems supporting the process of diagnostics and the treatment of MI [6]. All the above-mentioned achievements have significantly increased access to a modern invasive treatment. 

In the available literature, it is possible to find many reports on the assessment of the death risk of patients with MI. The first attempts to develop a system for the assessment of the death risk date back to as early as the 1960s [7,8,9]. As new methods for the treatment of MI (fibrinolytic treatment and invasive treatment) are developed and launched, the systems for the assessment of the death risk evolve. Over recent years, many systems assessing the death risk of patients with MI, both from short-term (in-hospital risk) and long-term (in various periods of time after concluding hospital treatment) perspectives, have been developed. Some of those systems constitute clinical scales that can be applied in the case of hospitalized patients, which can be useful in the everyday medical practice of a cardiologist [10].

Applying the clinical scales of the assessment of the death risk of patients with MI is recommended by the European Society of Cardiology (ESC) in its latest guidelines relevant to the treatment of MI. Currently, due to its high discriminative performance value, the GRACE score is the basic score recommended in the assessment of a patient with NSTEMI (non-ST-elevation myocardial infarction) [11]. In the ESC guidelines, it received a grade of class IIa, level of evidence (LOE): B recommendation [12]. Risk assessment scores are valuable tools for the physician, supporting the process of making clinical decisions, pointing to the optimal pharmacotherapy, helping to determine the duration of hospitalization, and facilitating the selection of post-hospital management strategies. By analyzing the systems of clinical assessment of the death risk in the case of patients with MI published in recent years, it is possible to observe that most of them rely on a similar set of variables relevant to initial clinical burdens, vital signs registered in the course of admission to hospital, the values of laboratory parameters, and MI’s manifestation in electrocardiography (ECG) records, which are assessed in the course of the first contact with patients. The analysis of these parameters allows the determination of the initial death risk of in-hospital patients admitted to hospital with diagnosed MI. This premise is the basis of, among others, the GRACE score [11], PROACS [13], and the ACTION score [14]. The above-mentioned systems of assessment of risk do not take into consideration other factors, the presence of which are ascertained during the course of the diagnosis and treatment of a patient and which may definitely change the level of death risk. Such factors include the parameters connected with the performed PCA (percutaneous coronary angioplasty), the profile of a treated atherosclerotic lesion, and the effectiveness of invasive treatment (blood flow scale applicable to the coronary artery—TIMI) [15]. An instance of a risk scale taking into consideration the above-mentioned aspects is the EURO HEART STEMI PCI score [16], which, as part of the assessment of the in-hospital mortality of patients with MI, not only takes into account the initial risk relevant to a patient, but also updates the value of this risk on the basis of parameters connected with PCA. It is also worth taking into consideration the type of vascular access obtained. Currently, only one of the risk assessment systems, namely, the ALFA score [17], includes this variable as a factor modifying the death risk relevant to patients with MI. It ought to be borne in mind that all the above-mentioned systems of assessment of the in-hospital death risk relevant to MI are based on data collected from patients treated in the period from 2002 to 2013. Another issue that is not discussed more broadly and precisely in the context of assessment of the death risk of patients with MI is the risk resulting from the overall logistical aspects of the treatment process. This is not about time delays during the infarction treatment procedure, of which the effect on the effectiveness of treatment is well described [18], but about improper organization of the workplace, the impact of the time of day or night or the day of the week (weekday, public holidays) that the patient is admitted to hospital (or undergoes an invasive procedure), and, finally, the experience of the staff or their workload on a given day. 

Analysis of the above-mentioned information leads to a clear conclusion: it is necessary to assess the potential new risk factors relevant to the in-hospital death of patients with MI. Further, it seems that, due to the above-described dynamic development of invasive cardiology and contemporary pharmacotherapy, reassessment of the significance of selected established risk factors of the in-hospital death of patients with MI, ought to be undertaken, taking into consideration clinical data collected exclusively over the last few years. 

## 2. Aim of the Study

The objective of this study was to conduct an assessment of in-hospital mortality as well as a multi-aspect assessment of risk factors relevant to the in-hospital death of patients hospitalized because of acute MI. Apart from established death risk factors, particular attention was paid to the assessment of new, alternative potential death risk factors. Significant attention was paid to factors connected with the logistics of the treatment process of MI, as well as to those connected with the parameters of the conducted invasive treatment. The results of this analysis will allow, in the future, the design of a new, multi-center registry of acute coronary syndromes, which might serve as a database in developing a new, modern model for assessing the risk of in-hospital death of patients with MI. 

## 3. Materials and Methods

### 3.1. ACS GRU Registry

This analysis was based on a single-center, retrospective, observational registry of patients admitted to the Intensive Cardiological Care and Invasive Cardiology Department of the Regional Specialist Hospital (RSH) in Grudziadz (Poland) because of MI. This hospital is a multi-profile, highly specialized medical center offering emergency care in all fields of specialization. The hospital also offers round-the-clock medical services of the Emergency Department (ED) (with the possibility of receiving an emergency helicopter at any time of the day or night at the airport operating within the hospital), the Cardiac Surgery Department, and the Intensive Care Unit (ICU). The Intensive Cardiological Care and Invasive Cardiology Department (hereafter called the Cardiological Department) provide comprehensive diagnosis and treatment of patients with cardiac diseases. The CL within the structures of the Cardiology Department is open round the clock every day, offers emergency care, and consults with emergency medical services (EMS) with the application of transtelephonic electrocardiography systems. The hospital has an Inter-Department Heart Team, composed of a cardiologist, cardiac surgeon, and invasive cardiologist. The registry of acute coronary syndromes in the hospital in Grudziadz (ACS GRU) was established on the basis of a retrospective analysis of digital medical histories. The only criterion for an adult patient to be entered into the registry is that they have to be diagnosed as suffering from MI (in accordance with the current guidelines relevant to diagnosing MI) [19]. The registry does not provide exclusion criteria. The analysis was conducted on the entire medical documentation of patients with MI initially admitted to the Cardiology Department (as well as that of patients secondarily transferred to other departments in order to undergo further treatment, e.g., the Cardiac Surgery Department or ICU). The personal data of patients were rendered anonymous. The study population consisted of patients diagnosed with MI who:Were transported by EMS from home/a public place;Were transferred from another department of RSH Grudziadz to the Cardiology Department;Were transported from another hospital to the relevant department of RHS Grudziadz.

Patients entered into the registry displayed a full spectrum of acute coronary syndromes: unstable angina (UA), NSTEMI, and STEMI (ST-elevation myocardial infarction). Both those treated invasively and conservatively were included in the registry. The choice of the method of treatment of patients always rested with the on-duty cardiologist and was based on European guidelines relevant to the treatment of acute coronary syndromes published by the ESC [20,21,22]. The registry, having the form of a digital database, included the demographic data of the patients, previous medical history, vital signs collected during the first contact with the hospital staff, results of additional tests, data related to the invasive procedures, the occurrence of various types of serious adverse events (SAEs), and the applied pharmacological treatment. The registry also included data relevant to the logistical aspects of the treatment process of MI.

### 3.2. Data Definitions 

Sudden cardiac arrest (SCA) was defined as evaluated by EMS or hospital medical personnel, and patients either received attempts at external defibrillation or chest compression by organized EMS or hospital medical personnel, or were pulseless at the time of presentation. 

Vital signs (heart rate, blood pressure, saturation) were determined at the time of first presentation at the hospital with the application of a vital signs monitor, EarlyVue VS30 (Netherlands). Electrocardiograms at presentation were performed in the Emergency Department (ED) by Aspel AsARD Mr. GOLD (Poland) and interpreted locally by the on-duty cardiologist according to ESC guidelines [21]. 

Blood for laboratory tests was collected immediately after admission to hospital and was processed by applying analyzers: Sysmex XN-2000 (Japan), Siemens BCS XP (Germany), and Siemens Healthineers Atellica Solution (Germany). The estimated glomerular filtration rate (eGFR) was estimated by applying the MDRD (Modification of Diet in Renal Disease) formula [23]. The results relevant to particular laboratory parameters were finally presented together with their qualitative interpretation, determining them as one of the following in accordance with the scope of the standard level adopted by the local laboratory:Below the standard level;At the standard level;Above the standard level.

The standard levels of laboratory examinations selected for the purpose of analyzing the statistical regression are presented in Table 1. 

The left ventricular ejection fraction (LVEF) was estimated applying the Simpson biplane method [24] along with a transthoracic echocardiogram with GE Vivid S6 (Boston, MA, USA). In accordance with the guidelines of the ESC relevant to diagnosing heart failure, the following values of LVEF were differentiated: regular (≥50%), moderately lowered (41–49%), and lowered (≤40%) [25]. 

All the operators at the CL (altogether, 8 physicians) were a priori divided into 3 groups depending on experience (time since having received a certificate of a fully qualified operator in invasive cardiology): operators with little experience (<5 years), operators with moderate experience (5–10 years), and operators with much experience (>10 years). The personal data of operators were rendered anonymous. This study also included the assessment of the impact of the tiredness of an operator on the mortality of patients subjected to invasive procedures as part of the treatment of MI. For this purpose, the documentation recorded the sequence of procedures performed on a given day by individual operators. On the basis of the collected data, the median number of procedures performed per day by operators was determined (2 procedures within 24 h of duty). The relationship between the mortality of patients and the number of consecutive procedures performed by the operator on a given day was assessed in two sub-groups (reference group, i.e., the group of patients whose procedure was the first or second consecutive procedure for the operator on a given day vs. research group, i.e., the group of patients whose procedure was the third or subsequent procedure for the operator on a given day).

Cardiogenic shock was defined as a decrease in blood pressure (RR < 90/60 mmHg) accompanied by the symptoms of organ hypoperfusion. 

The degree of heart failure at presentation was evaluated applying the Killip–Kimball classification [26].

The degree of coronary flow was determined with the application of the TIMI scale [15]. To be found ineffective, PCA needed to meet the following criteria: Impossible to move a coronary guidewire behind the culprit lesion;Impossible to place a balloon/stent in the culprit lesion.

The analysis of the logistical aspects gave rise to a priori classification of the period of admission to the department/performing a coronographic examination or a PCA, dividing them into those performed on weekdays and those performed on public holidays. In the study center, a standard regular working day is 8:00 A.M.–2:00 P.M. on weekdays; thus, the following sub-periods were defined: from 8:00 A.M. to 2:00 P.M., regular working hours of all doctors and nursing personnel; from 2:00 P.M. to 10:00 P.M., duty-mode work during the day; from 10:00 P.M. to 8:00 A.M. of the following day, emergency admission during the night. On public holidays (emergency admission round the clock), the following sub-periods were defined: from 8:00 A.M. to 10:00 P.M., emergency admission during the day; from 10 P.M. to 8:00 A.M. of the following day, emergency admission during the night.

Cigarette smoking was classified as both active nicotinism and having been addicted to smoking in the past. Diabetes was defined as having been diagnosed as suffering from this disease (of either type), regardless of the method of treatment.

The application of antiplatelet treatment was defined as receiving one or two of the following drugs—aspirin, clopidogrel, ticagrelor, and prasugrel—for no less than a month prior to admission to the Cardiology Department with the diagnosis of MI. Similarly, the application of anti-coagulative treatment was defined as receiving one of the following drugs—warfarin, acenocoumarin, dabigatran, rivaroxaban, and apixaban—for no less than a month. 

### 3.3. Course of the Treatment Process in the Case of Patients with MI—Logistical Aspects

After having been admitted to the ED, a patient was, in every case, assessed by a cardiologist on duty, and classified as one of the following:Patient with STEMI requiring an immediate invasive strategy;Patient with NSTEMI requiring an immediate, early, or selective invasive strategy, in accordance with the guidelines [12,21,22].

The documentation contained information on the place from which a patient was brought to the ED: Home/public place (by an EMS team);Another hospital (transfer after a prior arrangement);Transfer from another department of the same hospital.

The rules for collaboration as well as the criteria for the application of a transtelephonic electrocardiography system are described in detail in the literature [6,27,28]. If it was decided that a patient required an immediate invasive treatment, the patient was transferred straight to the CL, by-passing the Cardiology Department. In the case of STEMI, after a decision to apply invasive treatment, each patient received, at the ED or on site, a standard antiplatelet treatment: a loading dose of 180 mg ticagrelor (300–600 mg clopidogrel in the case of it being contraindicated to use ticagrelor) and 300 mg aspirin. If an EMS team or another department from which a patient with STEMI was transferred did not have ticagrelor at their disposal, it was recommended to administer 600 mg clopidogrel as soon as possible; after having arrived at the ED, such a patient (unless contraindicated) immediately received 180 mg ticagrelor. In the case of NSTEMI, a decision on the time and place of administering a standard antiplatelet treatment was made by an on-duty physician of the Cardiology Department. The PCA and adjunctive pharmacological medication were performed according to the ESC guidelines. Each patient, prior to performing PCA, received a standard anti-coagulative treatment (bolus of unfractionated heparin: 70–100 IU/kg iv; 50–70 IU/kg iv if concomitant with Glycoprotein IIb/IIIa inhibitors). It is standard procedure to use the right radial access; in the case of problems with securing it, an operator personally chose another type of vascular access. Angiographic coronary artery calcifications were evaluated each time by the operator performing the procedure. Calcifications were most often defined as radiographic changes visible in the coronary artery view before contrast administration [29]. The scope of the procedure (coronary angioplasty only of the infarction-related artery (IRA), and any subsequent planned angioplasty of the remaining stenotic arteries or angioplasty of several stenotic arteries in the course of a single procedure) was determined, in every case, by the physician performing the procedure. Glycoprotein IIb/IIIa inhibitors were administered when the operating surgeon considered it necessary. Pre-dilatation of the coronary artery using a balloon (semi-compliant (SC) or non-compliant (NC)), use of an aspiration thrombectomy catheter, and post-dilatation with an NC balloon after stent deployment were not routine strategies; the decision to perform the above was made each time by the operator performing the procedure. If a coronary stent was implanted, ticagrelor (clopidogrel) was prescribed for 12 months, along with aspirin. Intravascular ultrasound (IVUS) imaging was not performed as a standard, and the relevant decision was made by the physician performing the procedure. When an invasive cardiologist decided that a patient needed to be qualified for coronary artery bypass grafting surgery (CABG), the Heart Team’s consultation took place immediately at the CL. Patients classified as in need of CABG, depending on the clinical condition, were transferred to the Cardiac Surgery Department, or to the Cardiology Department to await the procedure to be conducted in the former department. After the PCA, patients were routinely transferred to the Intensive Cardiological Unit of the Cardiology Department. In the case of unconscious patients who were mechanically ventilated after the end of an invasive treatment, they were transferred either to the ICU or (if there were no free beds there) to the Intensive Cardiological Unit of the Cardiology Department. After their state had been stabilized, and the patients were no longer intubated, they were re-transferred from the ICU to the Cardiology Department. During the course of hospitalization, ticagrelor was replaced with clopidogrel at the clear and express request of a few patients (due to their financial situation). During their stay at the Cardiological Department, patients received a typical pharmacological treatment applied in the acute phase of MI. Within a day spent in the Cardiology Department, a cardiac echo examination was performed in the majority of cases. The documentation included all patient transfers between the departmentsas well as changes in antiplatelet treatment. 

### 3.4. Study Population

The registry included a final total of 633 adult patients (35.39% of whom were females) with an average age of 67.95 years (±4.95) admitted to the Cardiological Department of RSH in Grudziadz because of MI (both STEMI and NSTEMI/UA) over 12 consecutive months (from 1 February 2019 to 31 January 2020). All the patients were of European ancestry. The personal data of the patients were retrospectively rendered anonymous and entered into the local digital registry of acute coronary syndromes (ACS GRU), on the basis of which the statistical analyses were performed. The baseline characteristics of the study population are presented in Table 2. 

### 3.5. Statistical Analysis 

As the end point of the study, the in-hospital death of patients as adopted, regardless of cause. Data are presented as mean ± standard deviation (median and interquartile range) for continuous variables, and as frequencies and percentages for categorical variables. Extreme values for continuous variables were set to outer limits based on clinical judgment. For the purpose of the statistical analysis assessing the risk factors of death, variables selected on the basis of the literature review (analysis of existing assessment models of the death risk of patients hospitalized because of MI) and the initial statistical analyses were selected (the unadjusted association between each candidate variable and in-hospital mortality was tested with the Mann–Whitney U test for continuous variables and a chi-square test for categorical variables).

In the next stage, regression analysis was conducted; this method is applied to modeling connections between two or more variables. A dependent variable (interpreted) in the form of the death of a patient was singled out. For the purpose of the analysis of interpreting factors, a univariate logistic regression model was applied, which rendered it possible to single out independent variables, exerting a significant influence upon the death of a patient. The threshold for statistical significance was established as a *p*-value of <0.05. The data are presented in the form of odds ratios (ORs) together with 95% confidence intervals (95% CIs). The statistical analysis was conducted with the application of Statistica ver. 13.1. The study was conducted with the consent of the Commission of Bioethics (No. 8/KB/2021). The design respected the principles of the Declaration of Helsinki [30] and the principles of good clinical practices. 

## 4. Results

### 4.1. In-Hospital Mortality

General in-hospital mortality in the study group of patients amounted to 7.27% (60.9% constituted deaths during the course of the stay at the Cardiology Department, 17.4% were deaths during the course of procedures performed at the CL, and 21.7% were deaths during the course of the stay in other hospital wards, i.e., the ICU and the Cardiac Surgery Department). Mortality amongst females amounted to 8.9%, while that amongst males amounted to 6.3% (*p* = 0.2333). Mortality amongst patients during the procedures performed at the CL amounted to 0.99% (altogether, six patients died there). Amongst patients with MI, 4.9% of the cases were treated conservatively, and 4.1% were treated with the application of cardiac surgery (CABG and the procedures performed because of complications related to the natural course of MI), while 91% were treated invasively. Mortality in the group of STEMI patients amounted to 9.42%, while that in the group of NSTEMI/UA patients amounted to 6.10% (*p* = 0.1243). Mortality amongst patients treated invasively amounted to 5.8%, while that amongst patients treated conservatively amounted to 37.9% (*p* < 0.0000). The mortality in the group of patients treated invasively (on whom primary PCA was performed) amounted to 5.9%, while that in the group of patients treated with CABG amounted to 3.85% (*p* = 0.6611). The average time of stay in the hospital amounted to 8.44 days (minimum: 1 day; maximum: 141 days). Amongst all the patients classified as requiring invasive treatment (on whom coronarography was performed), 91.04% were treated with primary PCA, 4.15% were classified as requiring CABG, and revascularization was not performed in 4.81% (because of a lack of their consent for invasive treatment or due to the absence of stenosis in coronary arteries). In the case of 4.82% (4.58% of the entire population) of the patients on whom coronarography was performed, no significant stenosis in coronary arteries was ascertained. Hybrid treatment (CABG and a subsequent PCA) was applied in the case of four patients (14.28% of those treated with the application of CABG). After primary PCA, amongst the patients classified as requiring further invasive treatment (second-stage treatment) during the course of their stay at the hospital, invasive treatment was performed in the case of 41% of the patients, and in the case of the remaining patients, subsequent procedures were planned in the regular course of treatment. In the case of three patients, it was indispensable to perform another (third) procedure (in one case, because of sub-acute thrombosis in a DES; in another case, because of complications in the course of the previous procedure; and in yet another case, in order to complete the process of revascularization). Of the 39 patients initially classified as requiring CABG, 26 patients underwent an operation (12 patients were classified as not requiring/unable to undergo cardiac surgical treatment or did not express their consent for further treatment, while 1 patient underwent PCA). Amongst the patients requiring intensive medical care and mechanical ventilation after primary PCA, 69.9% were immediately transferred from the CL to the ICU; the remaining patients were, due to logistic considerations, admitted to the Intensive Cardiological Unit of the Cardiological Department. A recurring MI related to sub-acute thrombosis in a stent was diagnosed in the case of two patients (0.36% of the patients treated with primary PCA).

### 4.2. The Risk Factors of In-Hospital Death of Patients with MI

A significant increase in the death risk was observed in the case of patients with STEMI and NSTEMI (in comparison with the group of patients with UA, the group whose death risk was potentially the lowest), while a significantly higher OR, amounting to 16.255 (*p* = 0.0000), was observed in the case of patients with STEMI. A significantly increased death risk was observed in the case of patients who were admitted to hospital after an SCA incident (OR = 3.046; *p* < 0.0000). As part of the analysis of the age of the patients, the highest death risk was recorded in the oldest age group members, i.e., 90 + and those between their 80th and 89th years of life. Amongst the initial burdens on patients, a statistically significant increase in the risk of in-hospital death was proven in the case of nicotinism, chronic kidney disease, and diabetes mellitus. Receiving antiplatelet treatment before MI statistically significantly reduced the risk of in-hospital death. Within the scope of the laboratory examinations selected during the patients’ admission to the hospital, a statistically significant increase in the death risk was confirmed in the case of an elevated (over the standard) level of potassium concentration (K+), a reduced (below the standard) level of sodium concentration (Na+), and altered renal parameters (creatinine concentration above the standard level, below-standard levels of eGFR). At the stage of hospitalization at the ED, an increased death risk was confirmed in the following cases: Systolic blood pressure (SBP) < 90 mmHg;Diastolic blood pressure (DSP) < 60 mmHg;Saturation < 90%;Second and third degrees in the Killip–Kimball classification;Out-of-hospital SCA;Cardiogenic shock;Requiring therapy with the use of a respirator;Suffering from AF (atrial fibrillation)/AFL (atrial flutter) as confirmed in an ECG recording upon admission;Admitted in the course of ongoing cardiopulmonary resuscitation (CRP).

At the stage of treatment at the CL, an increased death risk was confirmed in the following cases: Patients on whom the procedure was conducted with access through the femoral artery;Patients with diagnosed triple-vessel disease with or without an affected left main (LM) artery;Patients after an unsuccessful PCA;Patients who developed cardiogenic shock during the PCA procedure;Patients who presented SCA during the PCA procedure and required CPR;Patients requiring mechanical ventilation during the procedure (regardless of the cause).

Non-compliant (NC) balloon post-dilatation performed after the procedure of stent implantation into the coronary artery significantly reduced the patients’ risk of in-hospital death. A significantly increased death risk was confirmed in patients with left ventricular ejection fraction (LVEF) < 40%. As part of the performed analysis of the logistical factors connected with the treatment process applicable to MI, an increased death risk was confirmed in the following cases:Patients transferred from another department of the same hospital;PCAs performed on a weekday from 10 P.M. until 8 A.M.

The detailed results from the analysis of the influence of the univariate logistic regression model on specific variables of the death risk of patients are presented in Table 3.

## 5. Discussion

### 5.1. Assessment of In-Hospital Mortality

The general in-hospital mortality of 7.27% in the group of patients hospitalized because of MI (both STEMI and NSTEMI/UA) is better than the results of other studies. In a meta-analysis of patients with MI published in 2019, in-hospital mortality (regardless of the type of acute MI) in a group of 615,035 patients was estimated at 8.8% [31]. As part of the analysis of mortality in the sub-groups of patients with MI, a statistically significant correlation relevant to the frequency of deaths of patients was proven only in the case of the method of treatment. Deaths were more frequent in patients treated conservatively than in those treated invasively (35.48% vs. 5.81; *p* < 0.05). The comparison of in-hospital mortality rates amongst patients with STEMI in comparison with those with NSTEMI/UA is compatible with the general trend that short-term mortality in patients suffering from STEMI is higher than that in patients with NSTEMU/UA [32]. The frequency of the occurrence of MINOCA (MI with non-obstructive coronary arteries) in the registry is similar to that observed in other significantly more extensive registries [33]. The frequency of the occurrence of acute and sub-acute thrombosis in a stent [34] in the registry in question was assessed as amounting to 0.36%, which is lower than that observed in other earlier published studies (amounting to 0.8%) [35]. Assuming that the early stages of thrombosis in a stent are mainly connected with the procedural aspects of the procedure: stent undersizing, malapposition, presence of residual dissection, stent fracture, and impaired TIMI flow [36], as well as polymorphisms of genes responsible for the metabolism of clopidogrel [37], it is concluded that the decreased frequency of occurrence of sub-acute thrombosis in a stent observed in the ACS GRU registry may be connected with the common application of modern antiplatelet drugs, as well as the more frequent optimization of the results of PCA with the application of IVUS [38]. 

### 5.2. Assessment of Particular Risk Factors Relevant to the In-Hospital Mortality of Patients with Cardiac Infarction

As part of the discussion, new logistic variables, which have not been used thus far, as well as variables whose impact on in-hospital mortality in patients with MI differs from the analogous results obtained in previously published death risk assessment systems, will be subjected to a detailed analysis. 

#### 5.2.1. Stage of Treatment at the ED

In analyzing particular clinical variables exerting an influence upon short-term prognoses applicable to patients with MI, one ought to observe the compatibility of the majority of the obtained results with previous studies assessing the risk of in-hospital death. The following were confirmed: an increased death risk connected with old age and burdens such as diabetes, nicotinism, and chronic nephropathy, as well as an increased risk connected with a low systolic or diastolic blood pressure, low saturation, and the third or fourth degree in the Killip–Kimball classification, ascertained upon admission to the ED. At the stage of patients’ admission to the ED, a significantly greater death risk in patients with STEMI and NSTEMI was confirmed. It ought to be borne in mind that this assessment was performed in comparison with a reference group of patients with MI such as unstable angina pectoris. When comparing STEMI and NSTEMI/UA, a statistically significant increase in the death risk was obtained in the case of STEMI (OR = 1.18; *p* = 0.2863); these data are not included in Table 3. Similarly, the mortality in the group of patients with STEMI was not statistically significantly higher than in the group of patients with NSTEMI/UA. In most of the analyzed scales of risk, STEMI was a factor increasing the death risk; however, these scales are based on clinical data from the beginning of the 21st century. This discrepancy in the assessment of the risk relevant to patients with STEMI may result from the improved treatment effectiveness in this group of patients, common access to the methods of interventional cardiology (primary PCA), the application of new DESs, modern antiplatelet drugs, and the common application of telematic systems reducing the time required to implement appropriate invasive treatments. When analyzing the impact of the patient’s initial clinical burden, a variable whose impact on the risk of death clearly differs from previous observations (i.e., it has no effect on in-hospital mortality) indicates the patient’s burden of coronary disease (in the form of stable past myocardial infarction, past coronary intervention, and past CABG). This observation may also indicate a significant improvement in the quality of care and treatment of patients with stable coronary artery disease in recent years.

#### 5.2.2. Laboratory Examinations 

The results of laboratory studies showing the influence exerted by particular variables upon the in-hospital mortality is not incompatible with the trends hitherto presented. Doubts may arise due to the lack of a statistically significant influence exerted upon in-hospital mortality by the increased concentration of high-sensitivity troponin I (hs-TNI) and creatine kinase MB (CKMB). Increased concentrations of the markers of cardiac ischemia are a recognized risk factor for the death of patients with MI [39]. The obtained result may be explained in two ways. The first interpretation is connected with the fact that only the first markings performed most frequently at the stage of staying in the ED were included in the analysis. In the case of patients with STEMI who were admitted to hospital within 1 h from when they first started to experience pain, these markings were frequently at the standard level. The other cause of this phenomenon may be the lack of division of the increased values of hs-TNI according to the determined scopes of these values. In the analyses, the only included values were the binary ones: results within the scope of the standard level and above it. This issue may be the cause of the lack of influence (observed in the study) of the increased concentrations of glucose and anemia observed during the patients’ admission to hospital on the in-hospital mortality of patients with cardiac infarctions, regardless of the fact that both hyperglycemia [40] and anemia [41] have for years been recognized as factors increasing the death risk in the case of patients with MI.

#### 5.2.3. Stage of Treatment at the CL

When analyzing the stage of treatment of patients with MI at the CL, attention was directed to the remarkably high increase in the risk of in-hospital death for patients in whom vascular access was obtained through the femoral artery. While proving beyond any reasonable doubt that there is a positive influence of radial access on the prognoses and mortality of patients with MI [42], the observed increase in the risk connected with femoral access found in this study is incomparably higher than that in other studies [17,43]. It ought to be borne in mind that, in the case of patients in a bad clinical condition, such as patients in cardiogenic shock, mechanically ventilated patients, and patients whose clinical condition and ECG examination results may suggest a complicated and demanding (in technical terms) procedure, many operators decide to immediately attempt to obtain access through the femoral artery. This fact may, at least partly, explain such a significant increase in the death risk of patients on whom a PCA was performed with access through this artery. This observation is compatible with the phenomenon of the risk treatment paradox in the choice of transradial access [44]. An unexpected result was the significant increase in the death risk in the case of patients whose MI was a result of restenosis in a DES. In the literature, there are no data relevant to the risk of in-hospital death in the case of patients with MI in the course of restenosis in a DES, and this variable itself has never been included in any clinical system of assessment of the death risk. One of the few studies related to the evaluation of the risk of death in patients with MI in the course of restenosis in DESs confirmed that the most frequent clinical presentation of restenosis in a DES is acute coronary syndrome, and it also reported significantly worse results in a one-year observation of a group of patients with MI in the course of restenosis in a DES in comparison with a group of patients where restenosis in a DES was not connected with MI [45]. Another interesting observation is the reduced death risk in the case of patients with NC balloon post-dilatation after stent deployment. To date, post-dilatation with an NC balloon after stent implantation in the setting of primary PCA has been a controversial and unclear issue [46]. In the literature, it is possible to find reports relevant to long-term benefits connected with stent post-dilatation during primary PCA in the course of STEMI [47]; however, there are no similar reports relevant to the risk of in-hospital death. In turn, the influence of such recognized death risk factors on in-hospital mortality, as indicated by the values of the TIMI flow scale, has not been confirmed. A significant increase in the death risk was confirmed in the case of patients in whom a number of serious adverse effects (SAEs) occurred during their PCA procedure, such as circulation arrest with subsequent CPR, respiratory insufficiency requiring intubation of the patient and subsequent therapy with the use of a respirator, and cardiogenic shock. In this dissertation, the following definition of an SAE presented by the Harvard Medical Practice Study is adopted: “an unintended injury or complication that results in disability at the time of discharge, death or prolonged hospital stay caused by health care management rather than by the patient’s underlying disease process” [48]. Taking into consideration the above-mentioned data, it is not possible, however, to presume that the occurrence of a given SAE (and a subsequent elevated death risk) is directly connected with a non-optimal result of the PCA procedure or the degree of complexity of the procedure. This correlation seems to be more complex, and the occurrence of an adverse effect is probably an autonomous and independent predictor of patient death. Further, it is possible to conclude a connection between the development of technologies applied in invasive cardiological procedures and the significance of recognized risk factors associated with an unfavorable anatomy of atherosclerotic lesions or a technically complicated course of the PCA procedure, while the importance of the occurrence of SAEs is even greater. There is a balance between the potential advantages and disadvantages associated with the use of modern medical technologies: on the one hand, the use of modern technology improves the results of treatment and, to some extent, makes us independent of many classical factors that increase patients’ risk of death; on the other hand, the use of modern and aggressive methods may result in a higher incidence of SAEs such as coronary artery dissection, aortic dissection, and coronary artery perforation, which may ultimately lead to cardiac arrest during the procedure.

#### 5.2.4. Logistical Factors

The observed connection between an increased death risk and the occurrence of SAEs makes it evident that it is necessary to pay attention to the assessment of the influence of different patient-independent factors of the death risk, which are certainly constituted by the logistical aspects of the treatment process. One of the most interesting conclusions from the study is from the analysis of the in-hospital logistical variables connected with the treatment process for MI. The patients transferred from another department of the same hospital (to which they had been admitted because of a medical problem other than MI) to a study center with a diagnosis of MI manifested a significantly greater death risk in comparison with patients who were admitted to hospital from home or a public place. It is comparatively easy to explain this finding by the significantly greater clinical burden and co-morbidities of those patients, which had a direct impact on the increased death risk. Aspects connected with the influence of the time of MI patients’ admission to hospital on their prognosis as well as the causes of these differences have been raised many times in the literature [49]. Conclusions from these studies remain unclear and mutually contradictory. Recent publications present evidence supporting the claim that there is a negative influence of out-of-hours admissions on the in-hospital mortality of patients with MI [31], while others confirm the lack of such an influence [50]. This study does not allow us to conclusively indicate the connection between out-of-hours admissions of MI patients and increased in-hospital mortality. Another aspect is the connection between the risk of in-hospital death of patients and out-of-hours primary PCA. In the literature, it is possible to find studies based on observations from the period 1997–2004 assessing this aspect in the case of patients exclusively with STEMI [51,52], which confirmed the increased in-hospital mortality of patients with STEMI treated with out-of-hours primary PCA at the CL (including at night). This study is based on more recently hospitalized patients with the entire spectrum of acute coronary syndromes, and indicates a higher risk of in-hospital death in the case of patients on whom primary PCA was performed on a weekday, between 2 P.M. and 10 P.M., and a significantly higher death risk in the case of procedures conducted on weekdays after 10 P.M. No similar correlation relevant to the times of performing primary PCAs was confirmed in reference to the procedures performed on public holidays. The above-mentioned findings may be explained as a result of the fact that, on a weekday, an operator always performs a few planned procedures (in the course of a stable coronary heart disease) from the time they start work (approximately 8 A.M.). In connection with the above, a primary PCA performed due to MI after 2 P.M. on a weekday always constitutes quite an addition to an operator’s workload and is usually the fourth procedure performed by the operator on a given day. In the case of procedures performed at night on weekdays, there is another burden; namely, that of tiredness and sleep deprivation. On public holidays, the number of procedures performed by an operator during a 24 h duty shift usually varies from 0 to 3, which, in comparison with a weekday (3–8 procedures in a 24 h period), is definitely less of a workload, and allows for an appropriate rest between subsequent procedures, which may exert a direct influence on the result of the procedure, and an indirect influence on patients’ risk of in-hospital death. The above-mentioned observations may be of particular significance in reference to the invasive treatment of NSTEMI; in this case, if an early invasive strategy is chosen [12], the primary PCA may be performed within the next 24 h. This allows for a shift in the time of performing the primary PCA until the period of regular working hours of the CL, or for making an appropriate change in the order of procedures planned for a day. The analysis also included the connection between the experience of an operator performing the PCA and the indicator of the in-hospital mortality of the patients operated on by the operator in question. The experience of an operator was measured by the number of years spent working as a fully qualified operator in the field of invasive cardiology. Such a methodological approach did not render it possible to obtain statistically significant correlations. Perhaps, a better choice would have been to change the criteria on the basis of which experience is assessed in the case of particular operators. One of the practical solutions is the assessment of the experience of an operator with the adopted unit of the number of PCAs performed annually. In the literature, it is possible to find reports confirming the connection between a superior experience of an operator (measured by the number of annually performed PCAs) and reduced in-hospital mortality of patients [53]. In connection with this, repeated analysis of data relevant to the mortality of patients was attempted, this time taking into consideration the experience of an operator measured by the number of performed procedures; however, after a few years, it was found to be impossible to obtain credible data relevant to the number of performed procedures by particular operators in 2019. Another interesting aspect that was not analyzed broadly in the past was the influence exerted by the tiredness of an operator (measured by the number of procedures performed on a given day) on the in-hospital mortality of patients undergoing PCA. The statistical analysis did not confirm an influence exerted by the tiredness of an operator on the in-hospital mortality of patients undergoing primary PCA (mortality in the group of patients on whom the procedure was performed as the third or subsequent procedure on the same day by the same physician was not higher than that in the group of patients in whose case the procedure was the first or second procedure on a given day for an operator). It ought to be borne in mind that these results are relevant to the death risk of patients during the course of the entire period of hospitalization rather than the death risk of PCA at the CL. 

## 6. Study Limitations

When analyzing the above results, the retrospective and observational nature of this study should be taken into account. On the one hand, such studies better reflect the reality of modern hospital treatment and do not carry the risk of data manipulation, and the research population is a direct reflection of the patient population; on the other hand, there is a risk of incomplete clinical data that may distort the final results of the analyses. Another consequence of this fact is the significant diversity of some data, rendering it impossible to analyze them statistically (an instance of this is the duration of the stay at the ED, ranging from a few minutes to between ten and twenty hours, depending on the type of myocardial infarction—STEMI vs. NSTEMI/UA). Another limitation is the relatively small headcount of the study population, which may limit possibilities for statistical analyses in sub-groups. It also ought to be borne in mind that the study was conducted in 2019 when the 2015 ESC NSTEMI guidelines were still in force [20]. Even though the difference between those guidelines and the currently valid ones is not significant, they are still not identical [12].

## 7. Conclusions

The 21st century has been a period of dynamic development in medical technology. This has had a noticeably substantial influence on the results achieved thus far in the treatment of MI. This influence contributes to each stage of the treatment for MI, from changes within the scope of the initial clinical profile of patients with MI, to changes within the scope of significance of particular factors of the death risk, including reducing the rate of in-hospital mortality. As a result of common access to contemporary medical care systems and their constant development as well as modern pharmacotherapy and the development of medical technology, risk factors that were once perceived to cause a higher risk of in-hospital death in patients with MI, such as angina pectoris, MI, PCA, or CABG in the past medical history, have lost their importance completely. Unfortunately, the contemporary development of technology also has its drawbacks. This study suggests a greater influence of variables such as SAEs and logistics factors on in-hospital mortality, which were previously not universally applied in the systems of assessment of risk. Currently, it is possible to imagine a new model of the assessment of risk relevant to the treatment of MI, in which logistical aspects constitute a sui generis natural environment of the entire treatment process of MI, potentially exerting an influence on the death risk at each of the stages of this treatment, whereas SAEs are a completely new axis of this model. It is also becoming clear that further studies on a significantly larger group of patients with MI are needed to assess the significance of particular SAEs, both those occurring in the course invasive treatment at the CL, and those observed in the course of further hospitalization in the cardiology department.

## Figures and Tables

**Table 1 ijerph-20-03603-t001:** Standard levels of selected laboratory parameters.

Laboratory Parameter	Standard Level	
White blood cells	W: 3.98–10.04 × 10^3^/µL	
	M: 4.23–9.07 × 10^3^/µL	
Hemoglobin	W: 11.2–15.7 g/dL	
	M: 13.70–17.50 g/dL	
Na+ (sodium)	135–145 mmol/L	
K+ (potassium)	3.7–4.9 mmol/L	
Urea	17–43 mg/dL	
Creatinine	0.67–1.17 mg/dL	
eGFR (MDRD)	>60 mL/min/1.73 m^2^	
Glucose	60–99.00 mg/dL	
hs-TNI	<58.0 ng/L (99th percentile of URL)	(RL)
CKMB	<5 ng/ml	

Abbreviations: hs-TNI—high-sensitivity troponin I; URL—upper reference limit; eGFR—estimated glomerular filtration rate; MDRD—Modification of Diet in Renal Disease; CKMB—creatine kinase MB.

**Table 2 ijerph-20-03603-t002:** Baseline characteristics of study population.

Variable	Females (*n* = 224)	Males (*n* = 409)
Age (years)	71.52 ± 11.70	66 ± 10.66
Body mass (kg)	74.81 ± 17.63	86.26 ± 18.20
Height (cm)	160.89 ± 6.12	172.51 ± 7.52
BMI	28.92 ± 6.66	28.97 ± 5.73
STEMI (%)	37.5%	33.9%
NSTEMI/UA (%)	62.5%	66.1%
State after SCA prior to hospitalization (%)	3.12%	3.17%
Arterial hypertension (%)	75%	68.95%
Coronary heart disease (%)	35.71%	38.14%
Chronic heart failure (%)	11.16%	14.18%
Diabetes mellitus	42.86%	32.76%
Hyperlipidemia (%)	42.86%	36.67%
Chronic kidney disease (%)	8.93%	9.53%
Dialysis treatment (%)	2.23%	0.98%
COPD/asthma (%)	9.82%	9.53%
Chronic or paroxysmal AF/AFL	12.05%	9.78%
Gout (%)	3.12%	4.89%
Stroke (%)	7.59%	8,80%
Antiplatelet treatment (%)	25.45%	28.12%
Anti-coagulative treatment (%)	9.82%	8.56%

Abbreviations: BMI—body mass index; STEMI—ST-elevation myocardial infarction; NSTEMI—non-ST-elevation myocardial infarction; UA—unstable angina; SCA—sudden cardiac arrest; COPD—chronic obstructive pulmonary disease; AF—atrial fibrillation; AFL—atrial flutter.

**Table 3 ijerph-20-03603-t003:** Analysis of the univariate logistic regression model’s influence on specific variables of the patients’ risk of in-hospital death.

Variable	OR	95% CI	*p*
Spectrum of MI UA (reference group)			
STEMI	16.255	2.788–34987	0.0000
NSTEMI	12.791	3.918–41.763	0.0000
SCA prior to admission	3.046	1.951–4.754	0.0000
STEMI of the anterior wall of heart (different location of MI—reference group)	1.411	0.896–2.223	0.1373
Age <40 years (reference group)			
40–49 years	0.555	0.094–3.266	0.5154
50–59 years	0.364	0.137–0.969	0.0431
60–69 years	0.493	0.237–1.026	0.0587
70–79 years	0.945	0.466–1.915	0.8755
80–89 years	1.779	0.900–3.514	0.0973
>90 years	3.958	1.159–13.510	0.0281
Medical history Nicotinism	1.890	1.280–2.792	0.0014
Antiplatelet treatment in the past medical history	0.617	0.398–0.956	0.0307
Anti-coagulative treatment in the past medical history	1.120	0.689–1.820	0.6470
Chronic kidney disease	1.487	0.990–2.235	0.0561
Prior stable coronary disease	0.943	0.689–1.292	0.7159
Prior MI	0.983	0.705–1.372	0.9197
Prior PCA	0.731	0.494–1.081	0.1164
Prior CABG	0.511	0.188–1.395	0.1895
Diabetes mellitus	1.722	1.265–2.343	0.0005
Laboratory parameters			
White blood cells (WBCs) Standard level (reference group)			
Below the standard level	1.992	0.474–8.367	0.3467
Above the standard level	1.455	0.668–3.165	0.3449
Hemoglobin Standard level (reference group)			
Below the standard level	1.606	0.751–3.437	0.2221
Above the standard level	0.611	0.158–2.357	0.4742
Potassium (K+) Standard level (reference group)			
Below the standard level	0.827	0.456–1.498	0.5301
Above the standard level	1.921	1.041–3.543	0.0366
Urea Standard level (reference group)			
Below the standard level	0.790	0.203–3.073	0.7337
Above the standard level	1.691	0.808–3.542	0.1634
Creatinine Standard level (reference group)			
Below the standard level	0.453	0.173–1.184	0.1061
Above the standard level	3.238	1.838–5.703	0.0000
eGFR (MDRD) ≥90 mL/min/1.73 m^2^ (reference group)			
60–89 mL/min/1.73 m^2^	0.501	0.277–0.908	0.0226
30–59 mL/min/1.73 m^2^	1.642	0.963–2.802	0.0687
15–29 mL/min/1.73 m^2^	4.092	1.867–8.970	0.0004
<15 mL/min/1.73 m^2^	1.420	0.515–3.915	0.4985
Glucose <90 mg/dL (reference group)			
90–126 mg/dL	0.186	0.068–0.511	0.0011
>126 mg/dL	1.294	0.675–2.497	0.4374
hs-TNI Standard level (reference group)			
Above the standard level	1.384	0.935–2.049	0.1045
CK MB Standard level (reference group)			
Above the standard level	1.137	0.839–1.541	0.4061
Vital signs upon admission			
Systolic blood pressure 90–140 mmHg (reference group)			
<90 mmHg	8.955	4.004–20.010	0.0000
>140 mmHg	0.286	0.157–0.519	0.0000
>200 mmHg	0.551	0.180–1.685	0.2959
Diastolic blood pressure 60–90 mmHg (reference group)			
<60 mmHg	4.168	2.512–6.914	0.0000
>90 mmHg	0.488	0.305–0.781	0.0028
Heart rate 60–100/min (reference group)			
<60/min	1.183	0.572–2.445	0.6504
>100/min	1.113	0.649–1.907	0.6977
Saturation >95% (reference group)			
<90%	2.064	1.163–3.660	0.0132
90–95%	1.146	0.697–1.887	0.5909
Killip–Kimball classification 1st degree (reference group)			
2nd degree	0.626	0.317–1.236	0.1775
3rd degree	2.263	1.158–4.522	0.0169
4th degree	5.289	2.505–11.167	0.0000
Admission to the ED during ongoing CPR	2.599	1.278–5.287	0.0084
Cardiogenic shock presented at the ED	4.196	2.534–6.948	0.0000
Therapy with the use of a respirator at the ED	3.725	2.408–5.672	0.0000
SCA with subsequent CPR at the ED	2.247	1.137–4.441	0.0198
3rd degree atrioventricular block presented at the ED	1.467	0.504–4.274	0.4825
AF/AFL	2.093	1.417–3.091	0.0002
Catheterization Laboratory			
Vascular access: Right radial artery (reference group)			
Left radial artery	3.051	0.645–14.442	0.1596
Right femoral artery	34.224	15.251–76.799	0.0000
Left femoral artery	52.476	13.738–200.443	0.0000
Result of coronarography study One-vessel disease (reference group)			
Double-vessel disease	0.486	0.217–1.085	0.0781
Multi-vessel disease with affected LM	0.939	0.526–1.678	0.8326
Multi-vessel disease without affected LM	2.967	1.632–5.394	0.0004
PCA on proximal LAD	1.804	1.236–2.634	0.0000
PCA on LM/proximal LAD	112.840	77.650–163.976	0.0000
Thrombus in IRA	0.720	0.347–1.494	0.3785
Application of glycoprotein IIb/IIIa inhibitors	1.115	0.647–1.921	0.6962
Coronary artery calcifications in IRA	1.438	0.943–2.192	0.0915
Restenosis in a DES in IRA	62.677	29.566–132.871	0.0000
Pre-PCA TIMI flow 3 (reference group)			
0	1.842	0.904–3.756	0.0926
1	2.327	0.798–6.786	0.1219
2	0.694	0.147–3.282	0.6451
Post-PCA TIMI flow 3 (reference group)			
0	2.247	0.660–7.647	0.1950
1	2.996	0.826–10.886	0.0950
2	1.152	0.444–2.993	0.7708
Number of arteries undergoing PCA 1 (reference group)			
2	0.505	0.182–1.402	0.1897
3	2.904	1.205–7.001	0.0176
Pre-dilatation with balloon (SC or NC)	1.576	0.760–3.269	0.2212
Post-dilatation with NC balloon after stent deployment	0.610	0.415–0.895	0.0115
PCA in bifurcation of the coronary arteries (application of kissing balloon technique or implantation of a stent into the collateral)	0.670	0.323–1.388	0.2810
Circulation arrest with subsequent CPR (CL stage)	4.402	2.426–7.988	0.0000
Cardiogenic shock (CL stage)	3.516	2.341–5.280	0.0000
Therapy with the use of a respirator (CL stage)	8.700	5.283–14.326	0.0000
Unsuccessful PCA	4.382	2.404–7.988	0.0000
LVEF Regular (≥50%) (reference group)			
Moderately reduced LVEF (41–49%)	1.291	0.681–2.450	0.4336
Reduced LVEF (≤40%)	1.985	1.157–3.404	0.0128
Logistical factors			
Admission from: Home/public place (reference group)			
Another hospital	0.604	0.304–1.200	0.1501
Another department	2.647	1.329–5.270	0.0056
Time of admission to hospital Weekday from 8 A.M. until 2 P.M. (reference group)			
Weekday from 2 P.M. until 10 P.M.	1.333	0.800–2.221	0.2696
Weekday after 10 P.M.	1.035	0.526–2.038	0.9200
Public holiday until 10 P.M.	0.976	0.513–1.855	0.9405
Public holiday after 10 P.M.	1.726	0.864–3.444	0.1219
Time of primary PCA Weekday from 8 A.M. until 2 P.M. (reference group)			
Weekday from 2 P.M. until 10 P.M.	1.102	0.609–1.993	0.7480
Weekday after 10 P.M.	2.540	1.202–5.367	0.0146
Public holiday until 10 P.M.	0.775	0.344–1.745	0.5380
Public holiday after 10 P.M.	0.930	0.285–3.034	0.9040
Subsequent procedure: ≤2 procedures (reference group)			
>2 procedures	1.005	0.718–1.408	0.9755

Unless otherwise noted, the reference group consisted of patients in whom the specific variable was not confirmed. Abbreviations: STEMI—ST-elevation myocardial infarction; NSTEMI—non-ST-elevation myocardial infarction; SCA—sudden cardiac arrest; MI—myocardial infarction; PCA—percutaneous coronary angioplasty; CABG—coronary artery bypass grafting; eGFR—estimated glomerular filtration rate; CKMB—creatine kinase MB; hs-TNI—high-sensitivity troponin I; CPR—cardiopulmonary resuscitation; ED—emergency department; AF—atrial fibrillation; AFL—atrial flutter; LM—left main; LAD—left anterior descending artery; NC—non-compliant; SC—semi-compliant; CL—catheterization laboratory; LVEF—left ventricular ejection fraction; IRA—infarction-related artery.

## Data Availability

The data presented in this study are available on request from the corresponding author.

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
