# Peer review of "Assessment of In-Hospital Mortality and Its Risk Factors in Patients with Myocardial Infarction Considering the Logistical Aspects of the Treatment Process—A Single-Center, Retrospective, Observational Study"

_ijerph, 2023, doi:10.3390/ijerph20043603_

Round 1

Reviewer 1 Report

the goal of the study is reasonably well described : it is a pilot study the aim of which is to indicate potential variables, which, in the future, could be applied to develop a new model of the assessment of the risk of in-hospital death with with particular emphasis on the logistical aspects of hospital care.

OK, This is an ambitious and very valuable task. We need such registry based studies to enhance health organization. It is clear that the authors have done a thorough analysis of their data and demonstrate their mastery. I also understand that in order to highlight new significant elements, one must examine all those that have been previously described. However, the 26 pages of the article appeared unbalanced to me on reading, at the expense of the intelligibility of potential new logistical factors. However, the 26 pages of the article appeared unbalanced to me on reading, at the expense of the intelligibility of potential new logistical factors. Please simplify the discussion and focus more on your goals, . Also, the last sentence of the abstract is difficult to understand, thank you for making it clearer. In this sentence you mentioned the experience of the operator, even though in the discussion you say that the methodology used did not allow a statistical study of this parameter. In the abstract and in lines 652 -660, I was not convinced by your use of the word paradox.

Why, not talk more classically about the balance of advantages and disadvantages inherent in any technique?

A detail, the term Caucasian race is obsolete and is considered inappropriate in scientific and medical publication. The appropriate term is: of european ascendance

Reviewer 2 Report

Congratulations to the authors for the effort put into this study.

At the same time, I think that the study is very ambiguous. First of all, there is no logic in the characterization of the patients included in the study. There are too many and very different parameters that define the patients, but which were mixed up in the statistical part. Nowhere in the study is it mentioned what type of statistical processing was done, with what software. The conclusions of the study and the interpretation of the results is not a relevant one, which provides an answer to the problem initially discussed. It is generally a very ambiguous study, lacking clarity and leaving much to be desired in terms of expression and the English language used.

Round 2

Reviewer 1 Report

Good answers, your article is now in agreement with the quality of the data collected.

Continue your registry activity, which is very valuable.

Best regards

Reviewer 2 Report

I believe that the requirements regarding the correction and improvement of this article have been realized by the authors.